

# A comparative in-silico analysis of autophagy proteins in ciliates

Erhan Aslan[1,2], Nurçin Küçükoğlu[1,2] and Muhittin Arslanyolu[2]

[1] Graduate School of Science, Department of Molecular Biology, Anadolu University, Eskişehir, Turkey
[2] Laboratory of Molecular Biotechnology and Enzymology, Faculty of Science, Department of Biology, Anadolu University, Eskişehir, Turkey

## ABSTRACT

Autophagy serves as a turnover mechanism for the recycling of redundant and/or damaged macromolecules present in eukaryotic cells to re-use them under starvation conditions via a double-membrane structure known as autophagosome. A set of eukaryotic genes called autophagy-related genes (*ATGs*) orchestrate this highly elaborative process. The existence of these genes and the role they play in different eukaryotes are well-characterized. However, little is known of their role in some eukaryotes such as ciliates. Here, we report the computational analyses of *ATG* genes in five ciliate genomes to understand their diversity. Our results show that *Oxytricha trifallax* is the sole ciliate which has a conserved Atg12 conjugation system (Atg5-Atg12-Atg16). Interestingly, *Oxytricha* Atg16 protein includes WD repeats in addition to its N-terminal Atg16 domain as is the case in multicellular organisms. Additionally, phylogenetic analyses revealed that E2-like conjugating protein Atg10 is only present in *Tetrahymena thermophila*. We fail to find critical autophagy components Atg5, Atg7 and Atg8 in the parasitic ciliate *Ichthyophthirius multifiliis*. Contrary to previous reports, we also find that ciliate genomes do not encode typical Atg1 since all the candidate sequences lack an Atg1-specific C-terminal domain which is essential for Atg1 complex formation. Consistent with the absence of Atg1, ciliates also lack other members of the Atg1 complex. However, the presence of Atg6 in all ciliates examined here may rise the possibility that autophagosome formation could be operated through Atg6 in ciliates, since Atg6 has been shown as an alternative autophagy inducer. In conclusion, our results highlight that Atg proteins are partially conserved in ciliates. This may provide a better understanding for the autophagic destruction of the parental macronucleus, a developmental process also known as programmed nuclear death in ciliates.

# INTRODUCTION

Autophagy is a catabolic lysosomal pathway in which long-lived proteins, cytoplasmic materials, organelles and even intracellular bacteria are eliminated from the cytoplasm in a selective or nonselective manner. Different types of autophagic processes have been defined. Among them, macroautophagy (hereafter referred to as autophagy) requires a double-membrane vesicle called the autophagosome which encloses part of the

Corresponding author
Erhan Aslan, easlan@anadolu.edu.tr, erhanaslan26@gmail.com

cytoplasm to deliver material to the lysosomes (*Xie & Klionsky, 2007*). More than thirty five autophagy-related genes (*ATG*) were identified by genetic screens in the yeast *Saccharomyces cerevisiae* and further analyses then showed that these genes are highly conserved among eukaryotes. Some of these genes, which are referred to as core autophagy machinery (*Xie & Klionsky, 2007*), encode proteins that are responsible for autophagosome formation. These machinery members are categorized into three main groups; the Atg9 cycling system (Atg1 kinase complex, Atg2, Atg9, Atg13, Atg18, Atg27), the phosphatidylinositol 3-kinase (PI3K) complex (Atg6, Atg14, Vps15, Vps30, Vps38), and the ubiquitin-like protein conjugation system (Atg3, Atg4, Atg5, Atg7, Atg8, Atg10, Atg12, Atg16) (*Xie & Klionsky, 2007*; *Mizushima, Yoshimori & Ohsumi, 2011*; *Jiang et al., 2012*). Some of these proteins have particular significance due to their biological function. For instance, Atg1 complex activity is required for starvation-induced autophagy induction. Atg6, a member of the PI3K complex, has also been shown to be involved in autophagy induction in an Atg1-independent manner. In addition, its mammalian orthologue, Beclin-1 has been shown to be associated with many human diseases including different types of cancer and neurodegenerative diseases (*Jiang & Mizushima, 2014*). Atg8, on the other hand, is a marker protein responsible for both autophagosome and puncta formation, and localization of the *ATG8* gene product is used as a diagnostic assay for autophagy (*Mizushima & Yoshimori, 2007*). Though autophagic processes are well-defined in myriad eukaryotic models, little is known about this pathway in ciliates.

Ciliates provide unique features for cellular and molecular biology related studies. These eukaryotic microbes are best characterized by their binucleated genome architecture, known as nuclear dimorphism (*Orias, Cervantes & Hamilton, 2011*). There are two types of morphologically and genetically different nuclei in the same cytoplasm. The polyploid macronucleus (MAC) governs the cell phenotype by providing all necessary transcripts for biological functions and serves as soma. The diploid micronucleus (MIC) is transcriptionally inert and functions as a germline nucleus. Although there are some differences between species, ciliates have more or less similar life cycles. For instance *Ichthyophthirius multifiliis*, a parasitic ciliate, requires a fish species as a host for its life cycle (*Coyne et al., 2011*). Here we describe the life cycle of *Tetrahymena thermophila* as a representative ciliate which contains three distinctive stages; (i) growth, (ii) starvation and (iii) conjugation. (i) During vegetative growth cells proliferate by binary fission. The MIC divides by mitosis whereas the MAC divides amitotically without spindle formation. (ii) When cells are depleted of food, they enter the starvation stage. (iii) Conjugation is the only sexual developmental stage of *Tetrahymena,* in which exchange of genetic material occurs following meiosis. Conjugation is initiated by the physical interaction of two different mating-types of starved cells. In a related ciliate, *Paramecium tetraurelia*, in addition to conjugation, there is another way to reproduce sexually, called autogamy (self-fertilization), in which conjugative events take place in a single cell without pair formation. In *Tetrahymena,* the MIC undergoes meiosis to produce four haploid nuclei. One of them selectively migrates to the anterior of the cell and undergoes one round of mitosis. One of these mitotic products is exchanged between pairs, while the remaining three meiotic products are degraded. After the exchange, stationary and migratory nuclei

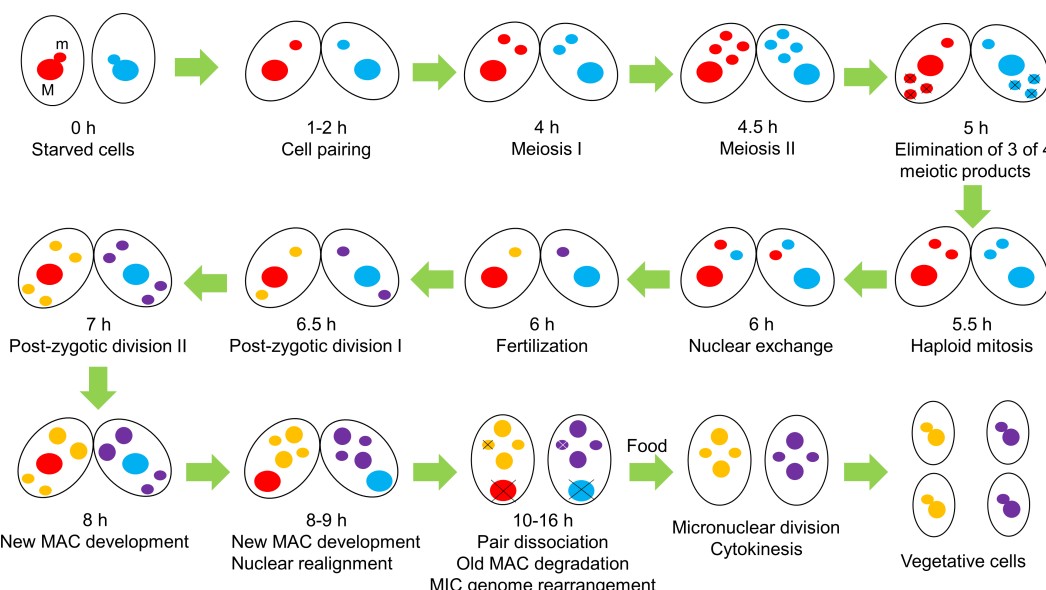

**Figure 1** **Life cycle and nuclear events during conjugation in *Tetrahymena thermophila*.** In the absence of sufficient nutrients, two cells of different mating types begin conjugation; the sexual reproduction stage of *Tetrahymena*. During conjugation, a number of nuclear events take place, including meiosis of the micronucleus (m) to yield four meiotic products, elimination of three of four of these products, haploid mitosis of the remaining meiotic product, nuclear exchange and fertilization, post-zygotic divisions of the fertilized nucleus, new macronucleus (M) development and elimination of the parental macronucleus in an apoptotic and autophagic fashion. When nutrients are available, exconjugants undergo cytokinesis and turn into vegetative cells. Approximate timing indicates time after mixing (hours) different mating types of starved cells (*Miao et al., 2009*).

fuse to form a diploid zygotic nucleus, which corresponds to the fertilization of sperm and eggs in metazoans. The zygotic nucleus then turns into four diploid products after two rounds of mitotic division. Two of these nuclei migrate to the anterior of the cell to differentiate into new MAC, whereas posteriorly positioned nuclei turn into MIC (*Orias, Cervantes & Hamilton, 2011*). Meanwhile, haploid meiotic products and poliploid parental MAC (old MAC) are degraded and eliminated from the cytoplasm in an apoptotic and autophagic-fashion by a genetically and developmentally programmed process, known as **P**rogrammed **N**uclear **D**eath (PND) (*Akematsu, Pearlman & Endoh, 2010*). Life cycle and nuclear events during conjugation in *Tetrahymena* are illustrated in Fig. 1.

PND in ciliates is a highly elaborate process which includes apoptotic and autophagic-like events. PND has been studied mostly with *Tetrahymena* (*Akematsu, Pearlman & Endoh, 2010*; *Akematsu & Endoh, 2010*; *Akematsu et al., 2014*), and molecular mechanisms of PND in other ciliates remain elusive. PND in *Tetrahymena* shares some phenotypic and biochemical resemblance to apoptosis or programmed cell death (PCD) in higher eukaryotes. For instance, chromatin condensation, loss of DNA integrity, genomic DNA degradation into high molecular weight DNA and oligonucleosome-size DNA fragmentation (DNA laddering) are considered as hallmarks of PCD, and all these are observed in PND of *Tetrahymena*. Consequently, several apoptotic molecules and organelles have been linked with PND in *Tetrahymena*. Mitochondria, for example,

include a variety of apoptogenic factors (*Van Gurp et al., 2003*). Kobayashi and Endoh investigated the role of mitochondria in PND in *Tetrahymena* using two different dyes, and discovered that mitochondria and the degenerating MAC were co-localized in autophagosomes (*Kobayashi & Endoh, 2005*). Moreover, they detected a mitochondria-driven nuclease activity functioning like mammalian EndoG during the DNA laddering process in *Tetrahymena* PND. Recently, it has been shown that Tmn1, a ciliate specific mitochondrial nuclease, is responsible for DNA laddering (*Osada et al., 2014*). Another apoptotic mitochondrial protein, AIF, is also involved in *Tetrahymena* PND. Knocking out the *AIF* gene in MAC caused a four hour delay in DNA fragmentation, but did not completely inhibit the PND (*Akematsu & Endoh, 2010*). Recently, we have detected increasing acidic DNase activity during late PND, implying a possible role for lysosomal DNase II (*Aslan & Arslanyolu, 2015*).

Recent studies in *Tetrahymena* also clearly revealed the involvement of Atg proteins in PND. Liu and Yao, for instance, showed that two GFP-tagged Atg8 proteins preferentially surround the degenerating parental MAC, but not newly-developing MAC in PND during conjugation. In addition, these proteins have distinct roles during starvation, and Δ*ATG8* cells show clear delay in nuclear degradation (*Liu & Yao, 2012*). Moreover, the role of class III phosphatidylinositol 3-kinase (PI3K) in *Tetrahymena* has been recently studied (*Akematsu et al., 2014*). It has been found that PI3K activity is required for autophagosome formation on the parental MAC. In the knockdown mutant cells, the parental MAC escaped from the lysosomal pathway. DNA fragmentation and final resorption of the parental nucleus are also found to be significantly blocked with the loss of PI3K activity. Although there is no report describing whether any Atg protein complexes exist in ciliates, the aforementioned studies show that Atg proteins have crucial roles in PND.

On the other hand, use of autophagic dyes shows that PND also involves an unusual form of autophagy which is different from yeast and mammalian autophagy. In yeast, autophagosome formation starts via accumulation of Atg proteins in a single dot-like site next to the vacuole, called the phagophore assembly site (PAS), and the mammalian cells have the counterpart of PAS (*Shibutani & Yoshimori, 2014*). However, in *Tetrahymena*, after nuclear condensation, the parental MAC changes its structure into an autophagosome membrane-like structure without accumulation of a pre-autophagosomal structure (PAS-like) in the cytoplasm (*Akematsu, Pearlman & Endoh, 2010*).

In this study, our aim is to use a comparative analysis to expand current knowledge of Atg proteins encoded in the genomes of model ciliates (Table 1). We find that, contrary to previous reports (*Rigden, Michels & Ginger, 2009*; *Duszenko et al., 2011*; *Liu & Yao, 2012*), all ciliate MAC genomes analyzed in this study lack a typical *ATG1* gene. We also discussed the presence and absence of Atg proteins in ciliates in terms of PND.

## MATERIALS & METHODS

### Sequence retrieval, BLAST and domain analyses

Five ciliate species, *Tetrahymena thermophila*, *Paramecium tetraurelia*, *Ichthyophthirius multifiliis*, *Oxytricha trifallax* and *Stylonychia lemnae* are analyzed in this study (Table 1).
**Table 1 Species analyzed in this study.**

| Species | Sequenced MAC genome | Prior experimental evidence of autophagy |
|---|---|---|
| *Tetrahymena thermophila* | *Eisen et al. (2006)* | Yes |
| *Paramecium tetraurelia* | *Aury et al. (2006)* | No |
| *Oxytricha trifallax* | *Swart et al. (2013)* | No |
| *Ichthyophthirius multifiliis* | *Coyne et al. (2011)* | No |
| *Stylonychia lemnae* | *Aeschlimann et al. (2014)* | No |

All these model ciliates have completed macronuclear genomes that are publicly available (*Aury et al., 2006*; *Eisen et al., 2006*; *Coyne et al., 2011*; *Swart et al., 2013*; *Aeschlimann et al., 2014*). Atg protein sequences were retrieved mainly from GenBank (http://www.ncbi.nlm.nih.gov/protein/) by probing *S.cerevisiae* Atg sequences against ciliates genomes in a PSI-BLAST analysis. When no similar sequences were obtained, other Atg proteins from humans, *P.pastoris* and *C.elegans* were also used as queries. BLAST analyses were initially performed in ciliates, and then the obtained candidate sequences were individually re-analyzed in species' genomes. An expected threshold (*E*-value) was set to 0.001 in all the BLAST-based analyses (*Jiang et al., 2012*; *Földvári-Nagy et al., 2014*). Unless otherwise stated, Atg complexes that we mentioned throughout our study come from yeast literature. The catalytic domains of candidate sequences were analyzed at CDD (*Marchler-Bauer et al., 2014*), HMMER (*Finn, Clements & Eddy, 2011*) and Pfam (*Finn et al., 2014*). The presence or absence of Atg proteins in ciliates are presented by the Dot-blot BLAST method (*Field, Coulson & Field, 2013*). Accession numbers of Atg proteins found in this study are given in Table S1.

## Phylogenetic analysis

Phylogenetic analyses were performed at online platform phylogeny.fr using the 'A la Carte' mode (*Dereeper et al., 2008*). Multiple alignment was conducted with MUSCLE followed by alignment curation with the 'remove positions with gaps' option. Unless otherwise stated maximum likelihood algorithm was chosen to construct the phylogenetic tree with default settings. For comparison, neighbor joining (NJ) and Bayesian trees were also calculated using default settings at phylogeny.fr. The constructed ultimate trees are displayed and annotated by iTOL (*Letunic & Bork, 2016*) at http://itol.embl.de/.

## mRNA expression analysis of *ATG* genes of *Tetrahymena* and *Paramecium*

mRNA expression data of *ATG* genes from *Tetrahymena* were obtained from the *Tetrahymena* Functional Genomics Database (TetraFGD) (*Miao et al., 2009*) at http://tfgd.ihb.ac.cn/. mRNA expression data of *ATG* genes from *Paramecium* were retrieved from the *Paramecium* Database (Paramecium DB) (*Arnaiz et al., 2010*) at http://paramecium.cgm.cnrs-gif.fr/.

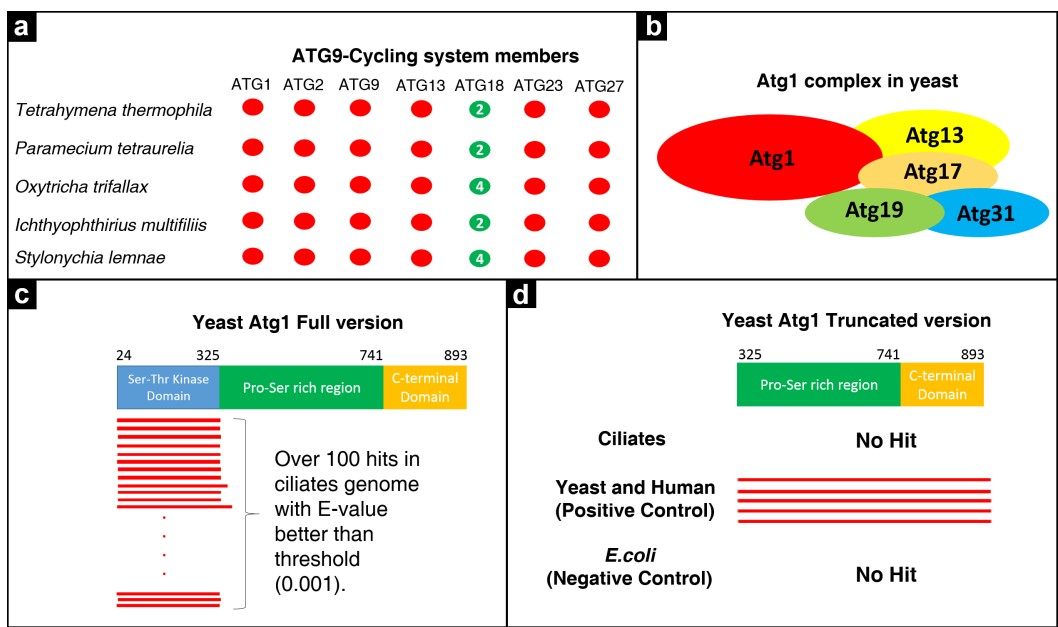

**Figure 2** **Analysis of Atg9-Cycling system members in ciliates.** (A) Distribution of system members among ciliates. Red circles denote the absence of indicated genes in the indicated species. Green circles show the presence of the indicated genes in the indicated species. A digit in the green circles refers to the number of significant hits. (B) Schematic depiction of Atg1 complex in yeast. Atg1 interacts with other components of the complex via its C-terminal domain. A diagram was constructed based on information described in *Földvári-Nagy et al., 2014*. (C) PSI-BLAST analysis of Atg1 in ciliate genomes using yeast Atg1. Significant hits only cover the N-terminal domain of the Atg1. (D) PSI-BLAST analysis of Atg1 in ciliate genomes using truncated version of yeast Atg1 which does not include the N-terminal domain. No significant hits were obtained in ciliates. Yeast and human genomes were also probed with the same truncated sequence and significant hits were observed (positive control). No hits were retrieved from the *E.coli* K12 genome as expected (negative control).

## RESULTS

### Analysis of Atg9-Cycling system members and lack of Atg1 protein in ciliates

The Atg9-Cycling system includes Atg9, Atg2, Atg18, Atg23, Atg27 and the Atg1 kinase complex (Atg1, Atg13, Atg17, Atg29 and Atg31) in yeast. Atg9 is the sole transmembrane protein in the core machinery and its activity is essential for autophagosome formation (*Xie & Klionsky, 2007*). It is known that Atg9 forms a complex with Atg2 and Atg18. As shown in Fig. 2, all ciliates analyzed in this study lack Atg9 and Atg2, but not Atg18 (Fig. 2A). We found two Atg18s in oligohymenophorean ciliates *Tetrahymena*, *Paramecium* and *Ichthyophthirius*. Stichotrichous ciliates *Oxytricha* and *Stylonychia lemnae* have four and five Atg18s encoded by their macronuclei, respectively (Fig. S1).

On the other hand, we were particularly interested in Atg1 in ciliates. Previous studies have reported a high number of *ATG1* genes in *Tetrahymena* and *Paramecium*, while yeast genomes have one and human encodes two (*ULK1-2*) (*Rigden, Michels & Ginger, 2009*; *Duszenko et al., 2011*). Atg1 is a kinase domain containing protein and its activity is essential for autophagy induction and autophagosome formation in yeast and mammals. It

contains three distinct domains; (i) a kinase domain at the N terminus, (ii) a proline-serine rich region at the center and (iii) a C-terminal domain. Both in yeast and mammals, Atg1 complex members (Atg13, Atg17, Atg29, Atg31 in the yeast Atg1 complex; Atg13, Atg101 and FIP200 in the mammalian ULK1 complex) require the C-terminal domain of Atg1 to interact and form the complex (*Chan & Tooze, 2009*) (Fig. 2A). Therefore, the presence of the C-terminal domain of Atg1 proteins appears crucial for function (*Chan et al., 2009*). Recently, *Földvári-Nagy et al. (2014)* analyzed non-unikont parasite genomes in terms of the presence or absence of Atg1 proteins. When they probed the complete yeast Atg1 sequence against 40 parasite species, they observed strong hits in return, but covering only the N terminal kinase domain. However, by using a truncated version of the yeast Atg1 sequence which lacks the N-terminal kinase domain, no significant hits were found. We applied the same experimental design in ciliate genomes and, when using yeast Atg1 as query, we also observed a large number of significant hits in ciliates with a coverage of only the N-terminal domain of the query (Fig. 2C). The truncated version of yeast and other Atg1 proteins, however, yielded no significant hits in ciliates (Fig. 2D). We conclude that these kinase domain containing proteins should not be considered as Atg1 candidates without further experimental analyses, since they all lack the important C-terminal domains. However, it is still an open question whether any of these kinase domain containing proteins have role(s) in autophagic processes in ciliates. Other components of this kinase complex (Atg1, Atg13, Atg17, Atg29 and Atg31) are also absent in ciliates. Collectively, these results may imply that ciliates might have an Atg1-independent pathway for autophagy induction and autophagosome formation.

## Analysis of PI3K complex members in ciliates

Vacuolar protein sorting 34 (*VPS34*) gene encodes phosphatidylinositol 3-phosphate (PI3P) and is required for autophagosome formation. In yeast, Vps34 forms two distinct complexes (*Kihara et al., 2001*). Only complex I (Vps34, Atg14, Atg6/Vps30 and Vps15) is found to be involved in autophagy (Fig. 3A). In complex II, Atg14 is replaced by Vps38, and this complex is involved in endosome to-Golgi trafficking, not in autophagy. Except for Atg14, the remaining PI3K complex members are found in ciliates. There is also experimental evidence for Vps15-Vps34 interaction in *Tetrahymena* (RE Pearlman, pers. comm., 2016). In addition, *Ichthyophthirius* lacks Vps15 (Fig. 3B, Fig. S2). The five ciliates discussed here have single Atg6 orthologue each. We generated two phylogenetic trees of Atg6 with different algorithms. In the both maximum likelihood and bayesian trees, ciliate members found to be clustered together (Figs. 3C–3D). The presence of Atg6 in ciliates seems important since Atg6 has the ability to induce autophagy in an Atg1 independent manner (*Földvári-Nagy et al., 2014*).

The mammalian counterpart of Atg6 is Beclin-1, and there are several non-yeast, mammalian Beclin-1 interacting proteins (*Kang et al., 2011*). We researched ciliate genomes to find the orthologues of these genes. We failed to find Ambra1, Dram, FIP200 or Atg101 in these ciliates. However, surprisingly, UVRAG (UV radiation resistance-associated gene), a Beclin-1 binding protein which promotes autophagosome formation by activating Beclin-1 complex (*Liang et al., 2006*; *Afzal et al., 2015*), is found only in *Oxytricha* among the ciliates

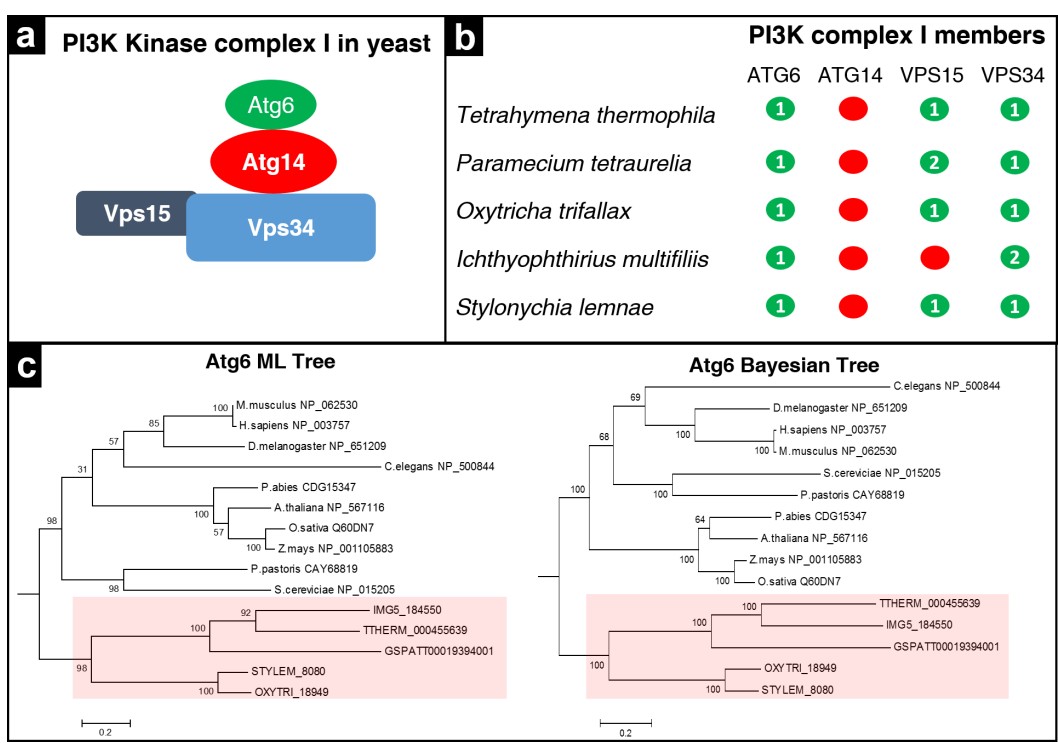

**Figure 3  Analysis of PI3K complex members in ciliates.** (A) Schematic model of PI3K Kinase complex in yeast according to (*Xie & Klionsky, 2007*). (B) Distribution of complex members among ciliates as in Fig. 2. (C) Maximum likelihood (ML) tree and Bayesian tree of Atg6. Trees were computed based on multiple alignment of Atg6 catalytic domains. Both trees show the same placement of critical nodes. Ciliate Atg6s were highlighted with pink color.

analyzed. Mammalian UVRAGs contain an Atg14 domain. However, we could not find any known domains in *Oxtricha*'s UVRAG in our domain analyses. We also found another non-yeast, Beclin-1 interacting autophagy protein, Vmp1 (Vacuole membrane protein 1), in all ciliates analyzed in this study (Table S1). Vmp1 is a transmembrane protein and is able to trigger autophagosome formation in mammalian cells. It has been shown that Vmp1 interacts with Beclin-1 through its C-terminal located Atg domain (*Ropolo et al., 2007*; *Molejon & Ropolo, 2013*). We found partial conservation between Atg domain of human Vmp1 and ciliate members (Fig. S3). This kind of interaction could compensate the absence of Atg1 in ciliates however, whether this is the case is not currently known.

## Analysis of ubiquitin-like conjugation system members in ciliates

In this group there are two conjugation systems; Atg8 and Atg12. The former includes Atg3, Atg4, Atg7 and Atg8 interactions. All of these Atg proteins are found in all ciliates despite the exceptional absence of *ATG7* and *ATG8* in *Ichthyophthirius* (Fig. 4). The expression level of Atg8 determines the size of the autophagosome (*Feng et al., 2014*). Atg8 is synthesized as a pro-enzyme and an arginine residue at its C-terminus is removed by Atg4, a cysteine protease, to expose a glycine residue which is necessary for interaction with E1-like enzyme Atg7, for its activation (*Mizushima & Yoshimori, 2007*). Therefore, the C-terminal glycine residue is critical for functional Atg8. We then analyzed Atg8 ciliate

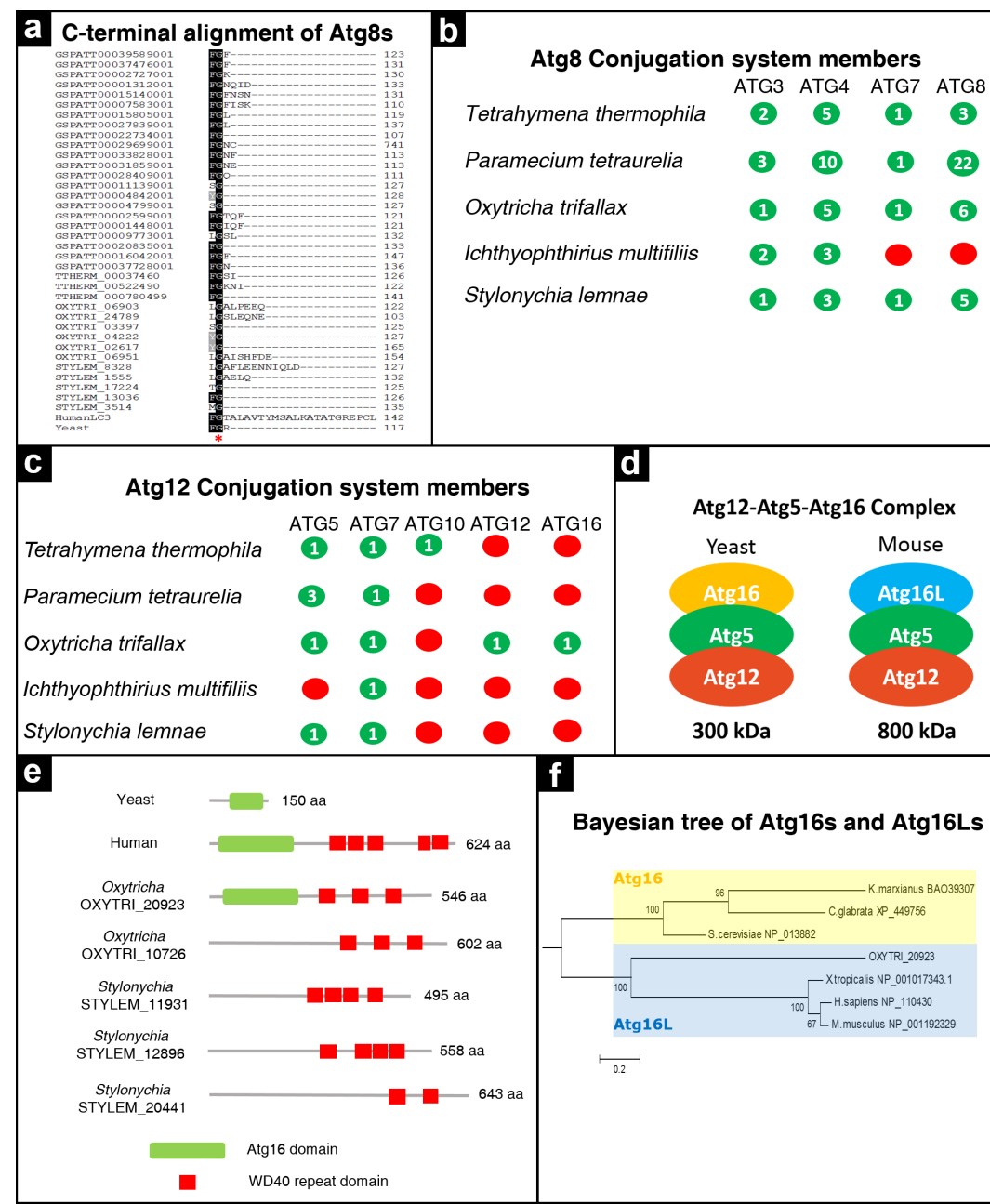

**Figure 4  Analysis of ubiquitin-like conjugation system members in ciliates.** (A) Multiple sequence alignment of C-terminal region of Atg8s. A red asterisk indicates the conserved glycine residue. Distribution of Atg8 (B) and Atg12 (C) conjugation system members among ciliates, as in Fig. 2. (D) Schematic depiction of Atg12-Atg5-Atg16 complex in yeast and mouse. (E) Domain analysis of Atg16 and Atg16L candidates in *Oxytricha* and *Stylonychia*. Accession numbers; yeast Atg16 (NP_013882) and human Atg16L (XP_005246139). (F) Bayesian tree of Atg16s and Atg16Ls. Tree was computed based on multiple alignment of Atg16 domains of yeast Atg16s and mammalian and reptile Atg16s. Tree shows that OXYTRI_20923 from *Oxytricha* is an Atg16L ortholog not Atg16.

candidate sequences for the presence of corresponding glycine residue (Fig. 4A). Three, five, six and 22 genes met the criteria in *Tetrahymena, Stylonychia, Oxytricha and Paramecium*, respectively. Interestingly, the *Ichthyophthirius* MAC genome lacks *ATG*8 and *ATG7*. Atg7 is represented by one gene in the remaining ciliate species, whereas other components of the Atg8 conjugation system differ greatly in numbers among species (Fig. S4). For example, *Paramecium* has more than 20 Atg8 and 10 Atg4 paralogues (Fig. 4B). This situation is explained by the fact that *Paramecium* has undergone whole genome duplications (WGD) during its evolutionary history (*Duszenko et al., 2011*). Other ciliates, whose evolutionary pasts do not include WGD, have relatively fewer Atg4 and Atg8.

The Atg12 conjugation system, on the other hand, involves the interaction of Atg5, Atg7, Atg10, Atg12 and Atg16. We found that Atg12 is present only in *Oxytricha*. Additionally, *Ichthyophthirius* lacks Atg5 (Fig. 4C, Fig. S5). Our BLAST analysis revealed that several genes in the MAC genomes of *Oxytricha* and *Stylonychia* have been annotated as Atg16 and Atg16-like (Atg16L). In yeast, Atg16 forms a 350 kDa complex with Atg12 and Atg5. In mice, however, this complex is much larger, 800 kDa (Fig. 4D). This difference is due to the long structure of mammalian Atg16 members. While yeast Atg16 protein contains one Atg16 domain, mammalian Atg16 proteins have seven WG repeat domains at their C-termini, in addition to the Atg16 domain (Fig. 4E). For this reason, mammalian Atg16 proteins are referred to as Atg16-like (Atg16L) (*Mizushima et al., 2003*). When we probed the ciliate genomes with yeast Atg16, no significant matches were found. BLAST analyses performed with mouse Atg16L resulted in a high number of significant hits in the ciliate species. However, the majority of these hits showed sequence similarity only to the WD40 domain of the query (Fig. 4E). On the other hand, some candidates had significant similarity with the query covering both the Atg16 and WD40 domains. Domain analyses for this Atg16 sequence showed that only the first hit (OXYTRI_20923) from *Oxytricha* seems to be a true Atg16L, but not Atg16 judging from the presence of both Atg16 and multiple WD40 domains (Fig. 4F, Fig. S6). Other significant hits, annotated as Atg16 and Atg16-1 isoforms, cannot be true Atg16, since they do not contain an Atg16 domain. Instead, they are simply WD40 repeat domains containing proteins (Fig. 4E). Based on these in silico observations, it seems promising to study the Atg12 conjugation system in *Oxytricha* since as it is the sole ciliate species carrying a conserved Atg12 conjugation system.

Previously Liu and Yao proposed TTHERM_00012980 from *Tetrahymena* as an Atg10 candidate (*Liu & Yao, 2012*). However, our domain analysis showed that this gene likely encodes an Atg3 protein (Fig. 5A). Cysteine residue in HPC motif in the catalytic domain and FLKC motif in the C-terminal domain are found in this sequence, as well as in other Atg3 candidates (Fig. 5B). During our BLAST analysis we could not find any Atg10-like proteins when Atg10 was used as a probe. However in Atg3 BLAST analysis, we found three candidate sequences, two of them possess Atg3-specific domain organization. The remaining candidate, on the other hand, contains only single catalytic domain (TTHERM_01016200) (Fig. 5A). Atg3 and Atg10 have "Autophagy_act_C" (PF03987) catalytic domain and catalytic sites of both proteins are similar (*Yamaguchi et al., 2012*). We therefore decided to compare this sequence with yeast and human Atg10 and found very low sequence homology although catalytic cysteine residues are conserved (Fig. 5C).

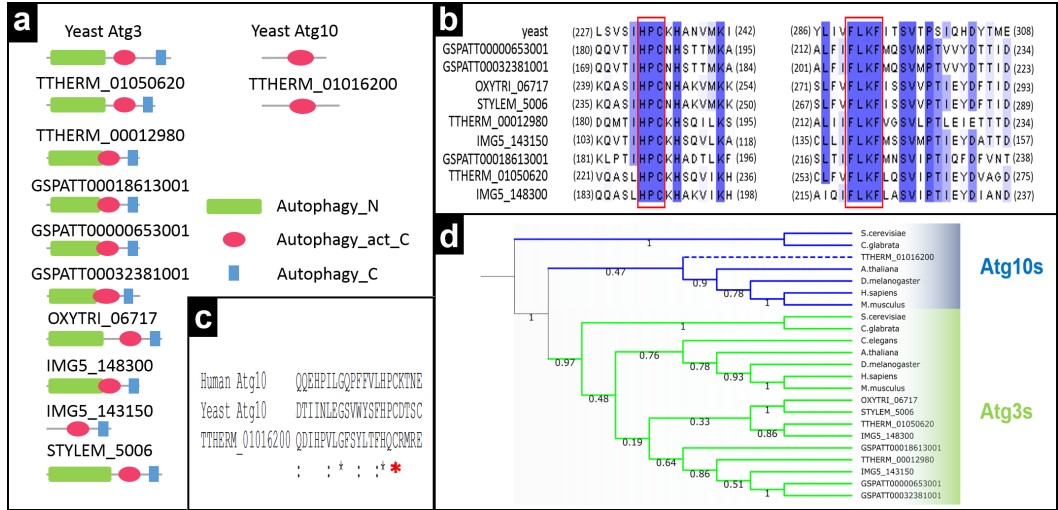

**Figure 5** *In silico* characterization of Atg3 and Atg10 in ciliates. (A) Domain analysis of Atg3 and Atg10. (B) Partial multiple alignment of Atg3s in ciliates. Atg3-specific conserved motifs were boxed with red rectangles. (C) Partial multiple alignment of the region containing catalytic cysteine (Red asterisk) residues of *Tetrahymena* Atg10-like protein with yeast and human Atg10 orthologues. (D) Neighborhood joining tree of Atg3s and Atg10s. Bootstrap values are given at each node (max 100 replicates). Dotted line indicates *Tetrahymena* Atg10-like protein.

In order to clarify whether this candidate is an Atg3 or Atg10, we computed a phylogenetic tree using Atg3-Atg10 sequences (Fig. 5D, Fig. S7). Atg3 and Atg10s formed two main clades as expected and TTHERM_01016200 was nested within the Atg10 clade. These phylogenetic results, along with domain analysis, suggest that this protein is likely an Atg10 despite its low sequence similarity with experimentally shown Atg10s. No Atg10 found in other ciliates.

## Phylogeny of Atg4s and Atg8s in ciliates

While the yeast genome encodes a single Atg4 and Atg8 proteins, there are multiple homologues of these proteins in mammals like ciliates. It has been shown that among the four Atg4 proteins in humans (Atg4a-d), Atg4b and Atg4c are able to fully restore autophagic deficiencies in *atg4Δ* yeast cells, as revealed by complementation experiments (*Marino et al., 2003*). In addition, Atg4b has the broadest capacity to use Atg8 proteins as substrate (*Li et al., 2011*). Mammalian Atg8 homologues are microtubule-associated protein 1 light chain 3 (LC3), $\gamma$-aminobutyric acid receptor-associated protein (GABARAP) and Golgi-associated ATPase enhancer of 16 kDa (GATE- 16) (*Slobodkin & Elazar, 2013*). In order to understand the phylogenetic relationships of Atg4s and Atg8s in ciliates with higher eukaryotes, maximum likelihood phylogenetic trees were constructed (Fig. 6). In the Atg4 phylogeny (Fig. 6A), two main clades were observed. The first clade (collapsed nodes, triangle) consists of only ciliate-specific Atg4 members. However, the other clade includes yeast Atg4, mammalian Atg4s and several Atg4 proteins from stichotrichous ciliates (except TTHERM_00622890, which belongs to *Tetrahymena*). We also constructed a phylogenetic tree for Atg8s (Fig. 6B). Since Atg8 and Atg12 are phylogenetically related, the tree was

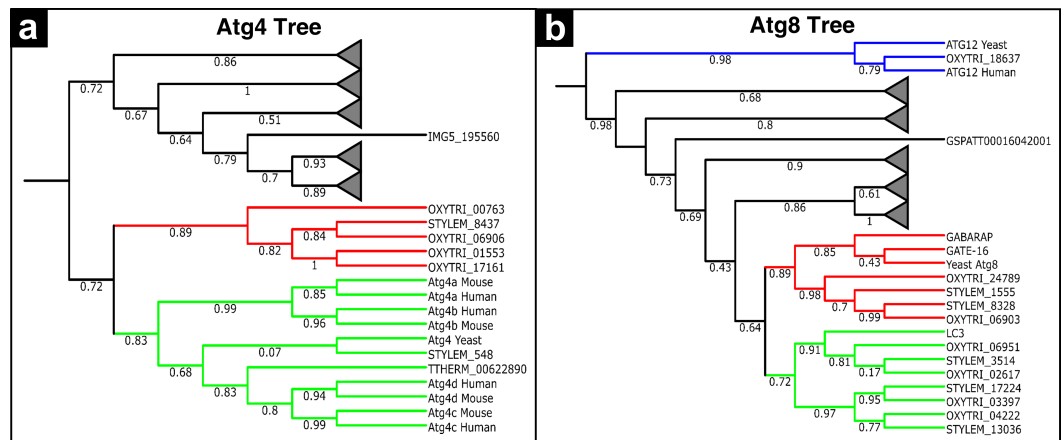

**Figure 6** **Phylogenetic analysis of ciliate Atg4s, and Atg8s.** Maximum likelihood trees were calculated for Atg4s (A) and Atg8s (B) as described in the Materials & Methods section. Bootstrap values are given at each node and branch lengths in the tree were ignored. Ciliate-specific clades were collapsed and shown as triangles.

rooted with a group of Atg12 proteins (*Williams et al., 2009*). Consistent with our BLAST and domain analyses, *Oxytricha* Atg12 (OXYTRI_18637) nested within the Atg12 clade, which further confirms that it is a real Atg12 orthologue. In addition to outgroup, the topology of the tree consists of four other main clades. While three of these clades include only members from *Paramecium* and *Paramecium + Tetrahymena*, the last remaining clade includes Atg8 members from *Oxytricha, Stylonychia*, yeast and humans. These results indicate that the Atg4s and Atg8s of stichotrichous ciliates are phylogenetically related to the Atg4s and Atg8s of higher eukaryotes.

## Complete absence of non-core Atg proteins in ciliates

In addition to core machinery members, several gene products are only required for non-selective and pathway-specific (selective) autophagy, while core proteins play a role in both selective and non-selective autophagy. For instance, an Atg17-Atg19-Atg31 complex is specifically required for non-selective autophagy in *Saccharomyces cerevisiae* (*Kabeya et al., 2009*). On the other hand, a set of proteins (Atg11, Atg19-21, Atg23-28, Atg30, Atg32 and Atg33) are required for selective autophagy, such as the Cvt pathway, pexophagy and mitophagy (*Lynch-Day & Klionsky, 2010*). We failed to identify any of these genes in ciliates, which indicates that ciliates only possess a part of the core *ATG* genes.

## mRNA expression analysis of *ATG* genes from *Tetrahymena* and *Paramecium*

In order to get an idea about the role of autophagy genes in the degradation of the parental MAC, we analyzed mRNA expression profiles of *ATG* genes during *Tetrahymena* conjugation and *Paramecium* autogamy. These are the only two ciliates in this study that have publicly available microarray data; therefore our comparative analysis is limited to these two ciliates (*Miao et al., 2009*; *Arnaiz et al., 2010*). We found that *ATG3*s and *ATG10* in *Tetrahymena* reach their maximum expression levels in conjugation with different

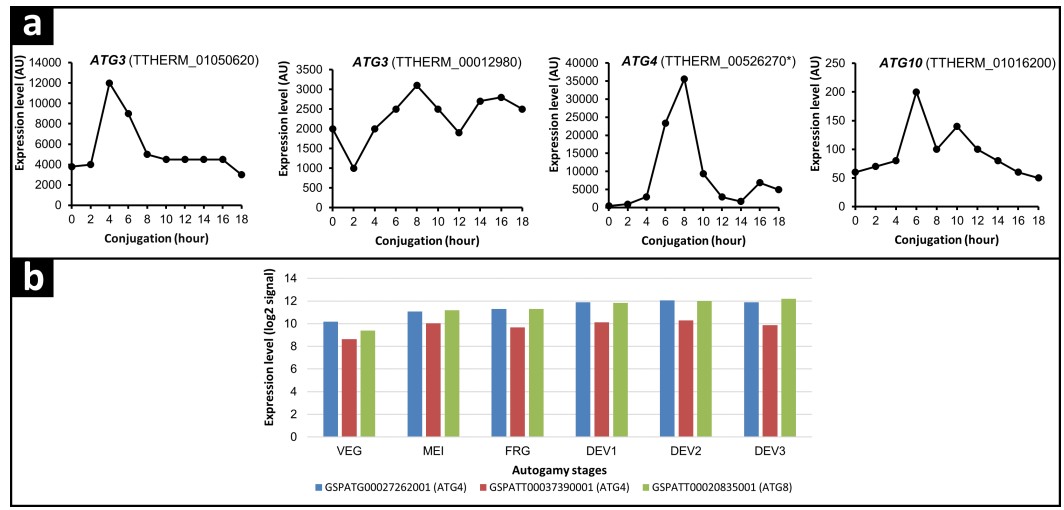

**Figure 7** **mRNA expression analysis of ATG genes upregulated during *Tetrahymena* conjugation and *Paramecium* autogamy.** (A) Data shows the mRNA expression profile of *ATG3*s, *ATG4* and *ATG10* during conjugation of *Tetrahymena*. The profiles are obtained from the *Tetrahymena* Functional Genomics Database-TetraFGD (http://tfgd.ihb.ac.cn/). Data represents the average of the results obtained by Gorowski, Miao and Pearlman Labs (*Miao et al., 2009*). *Note that TTHERM_00526270 contains two genes; one *COI3* (transcript id: gene_000015720), and the other, Atg4 (C54 peptidase domain, transcript id: gene_000006383) (*Woehrer et al., 2015*). We focused on the latter transcript in this study. (B) Data shows the mRNA expression profile of *ATG4s* and *ATG8* during autogamy of *Paramecium*. The profiles are obtained from the *Paramecium* Database (Paramecium DB) at http://paramecium.cgm.cnrs-gif.fr/. The time course of autogamy was characterized by nuclear morphology as follows: VEG; log-phase vegetative cells showing no signs of meiosis. MEI; beginning of macronuclear fragmentation and micronuclear meiosis. FRG; population in which about 50% of cells have a fragmented old macronucleus. DEV1; earliest stage at which a significant proportion of cells has visible macronuclear anlage.DEV2; majority of cells with macronuclear anlage. DEV3; population of cells ten hours after DEV2. Expression levels are shown as log2 values of raw measurements (*Arnaiz et al., 2010*).

timing. For instance, TTHERM_01050620 is expressed constitutively at a quite high level with a peak of expression at the initiation of starvation, reaching its peak expression at the fourth hour of conjugation, when meiosis II is completed. The expression level of TTHERM_00012980 is quite constant throughout the life cycle, but slightly higher at eight hours of conjugation in which PND also starts. The only Atg10-like protein in ciliates, TTHERM_01016200, is expressed at quite a low level and shows its maximum expression at six hours of conjugation, which coincides with nuclear exchange (Fig. 7A). Collectively, these results show that expression of *ATG3*s and *ATG10* in *Tetrahymena* are differentially regulated during the life cycle, depending on the nuclear stages. Since expression levels of these genes are higher between four to eight hours of conjugation, it is possible to postulate that TTHERM_01050620 (Atg3) and TTHERM_01016200 (Atg10) may be involved in the degradation of meiotic products and TTHERM_00012980 (Atg3) may take some roles in the parental MAC turnover during *Tetrahymena* conjugation. On the other hand, expression levels of *ATG3*s in *Paramecium* autogamy show no significant change throughout the life cycle.

Among the five *ATG4* genes, only TTHERM_00526270 shows conjugation-specific expression at eight hours when PND starts (Fig. 7A). Interestingly, two of ten *ATG* genes of *Paramecium* (GSPATG00027262001 and GSPATT00037390001) also show significant expression during autogamy at meiosis and parental MAC fragmentation stages (Fig. 7B). Therefore, we suggest that Atg4 may function during PND of ciliates.

*Liu & Yao (2012)* showed that *ATG8*s in *Tetrahymena* have conjugation-specific expression profiles covering the four to eight hours of conjugation in which the three meiotic products and parental MAC are eliminated. Among the twenty two *ATG8* genes of *Paramecium*, we found that only GSPATT00020835001 has significant expression during autogamy, not only at the meiosis and fragmentation stages, but also at the remaining stages of autogamy, such as development 1, 2 and 3 (Fig. 7B).

## DISCUSSION

In this study, we bioinformatically analyzed Atg proteins in ciliates. Our study is a continuing effort that deals with the evolutionary differentiation of Atg proteins in ciliates, since several groups have previously analyzed Atg proteins in protists. *Rigden, Michels & Ginger (2009)* reported the first thorough bioinformatics analysis in protists, including *Tetrahymena* and *Paramecium,* with evolutionary aspects of the presence and absence of *ATG* genes. *Duszenko et al. (2011)* also reviewed autophagy in protists in a similar approach to Ridgen et al. *Brennand et al. (2011)* advanced autophagy knowledge in parasitic protists, mainly for possible drug targeting as well as for the evaluation of the distribution of *ATG* genes. However, our current study possesses certain novelties, which differentiates this paper from previous work as discussed below.

In the aforementioned studies, authors have focused mainly on parasitic protists with least emphasis on free-living protists, such as *Tetrahymena* and *Paramecium*. Later on, additional macronuclear genomes of other model ciliates, *Oxytricha* (*Swart et al., 2013*), *Ichthyophthirius* (*Coyne et al., 2011*) and *Stylonychia* (*Aeschlimann et al., 2014*), were fully sequenced. Thus, it was essential to analyze Atg proteins in the light of newly-sequenced ciliate MAC genomes.

Remarkably, previous studies reported the presence of multiple *ATG1* genes in *Tetrahymena* and *Paramecium*. For example, Liu and Yao determined sixty candidate *ATG1* genes in *Tetrahymena* (*Liu & Yao, 2012*). However, as with others, they also used the complete Atg1 sequence in their BLAST analysis as a query. Atg1 is a kinase domain containing protein and, considering the fact that these kinds of proteins are common in ciliates (*Eisen et al., 2006*), it is therefore likely that these *ATG1* candidate genes, detected by Liu and Yao, may be false positive, since all these sequences lack a critical C-terminal domain, which is essential for the formation of the Atg1 complex (*Chan & Tooze, 2009*; *Földvári-Nagy et al., 2014*). However, we cannot exclude the possibility that these enzymes might have roles during PND and/or any autophagic-like processes in ciliates.

In agreement with previous studies, our results also show that several key Atg proteins, such as Atg1, Atg9 and Atg14, are absent, not only in *Tetrahymena* and *Paramecium*, but also in ciliates. This could be at least one of the predictable reasons why autophagosome

formation in PND of *Tetrahymena* differs from mammalian and yeast autophagy. Akematsu and colleagues (*2010*) showed the occurrence of autophagosome-like structures only on parental MAC, not on newly-developing MAC, without accumulation of PAS-like membrane structures. Their transmission electron microscopy analysis revealed that parental MAC does not possess a double-layered membrane derived from PAS, which is the case in typical macroautophagy (*Akematsu, Pearlman & Endoh, 2010*). In yeast, the Atg1 kinase complex is located at the PAS. Similarly, PAS is one of the localization sites of Atg9 in yeast (*Feng et al., 2014*). Atg14, on the other hand, is responsible for directing the PtdIns3K complex to the PAS (*Obara, Sekito & Ohsumi, 2006*). However, genes coding these proteins seem to be absent in ciliate MAC genomes. Considering the critical roles of these proteins in autophagosome formation, it would be reasonable to assume that a lack of canonical autophagosome formation during PND in *Tetrahymena* may be due to the absence of critical core machinery components, such as Atg1, Atg9 and Atg14. However, it is worth noting that some autophagosome-like structures have been observed in starved *Tetrahymena* (*Liu & Yao, 2012*) and *T.pyriformis* (*Nilsson, 1984*). However, autophagosome formation during starvation and/or conjugation in other ciliates still remains elusive.

Previously, it has been shown that genetic defects in *ATG8* and *VPS34* genes causes abnormal nuclear positioning during PND in *Tetrahymena* (*Liu & Yao, 2012*; *Akematsu et al., 2014*). In addition, Atg8-2 has been shown to be responsible for the selective marking of parental MAC for degradation in *Tetrahymena*. Interestingly, our analysis revealed that *ATG8* (also *ATG5, ATG7* and *VPS15* which are present in other ciliates) is not present in the *Ichthyophthirius* genome. Given that Atg8 is required for autophagosome formation and is used to monitor autophagy, the absence of Atg8 in *Ichthyophthirius* can raise the possibility that autophagy, in this parasitic ciliate, may not be operated in a canonical manner. Indeed, it is highly questionable whether *Ichthyophthirius* has sexual reproduction since conjugation in *Ichthyophthirius* has not been studied in depth to date. Although in a very recent study, population genetics data showed that this organism also reproduces sexually, but that conjugation-related genes in this ciliate are poorly-conserved compared to *Tetrahymena*, suggesting that mechanical differences between *Ichthyophthirius* and other ciliates may exist (*MacColl et al., 2015*). Therefore, currently it is difficult to speculate about the role of *ATG* genes during PND in *Ichthyophthirius*. However, further analyses on *ATG* genes in this parasitic ciliate may be helpful for possible drug targeting to combat with the white spot disease, caused by *Ichthyophthirius* in fresh water fish.

Comparative mRNA expression profile analysis showed that the expression of several *ATG* genes is up-regulated during *Tetrahymena* conjugation and *Paramecium* autogamy (Fig. 7). Experimental and statistical approaches are important in microarray studies, especially when deciphering microarray data that comes from two different organisms. For example, in a *Paramecium* microarray study, the authors performed TREAT analysis, a bioinformatics approach, to test significance relative to a fold-change threshold in microarray experiments (*Mccarthy & Smyth, 2009*; *Arnaiz et al., 2010*). According to this approach, we found only three genes with significant expression change in *Paramecium* autogamy (2 *ATG4* and single *ATG8*) (Fig. 7B). However, such a bioinformatics analysis

has not been performed in a *Tetrahymena* microarray study. For this reason, we subjectively determined the differentially expressed *ATG* genes during the conjugation of *Tetrahymena*. Objectively, it is still possible that statistically or subjectively significant expression changes may not be biologically meaningful. Likewise, a statistically insignificant expression profile may be biologically meaningful. To clarify this, further experimental evaluations are required.

## CONCLUSIONS

In this study, we bioinformatically analyzed Atg proteins in ciliates. Ciliates constitute as model organisms for plenty of molecular and cellular studies, including telomere biology, self-splicing introns and RNAi-guided genome shaping. Recent studies have shown that different types of apoptotic and autophagic mechanisms exist during the destruction of the parental nucleus in *Tetrahymena*. Starvation is not only a strong physiological inducer of autophagy, but also a physiological and developmental stage of ciliates. This provides a good opportunity to study crosstalk between apoptosis and autophagy. The presence of certain genes solely in *Oxytricha* (like *ATG12*, *ATG16L* and *UVRAG*) and the absence of certain critical genes in *Ichthyophthirius* (like *ATG5*, *ATG7* and *ATG8*) might open new platforms in the understanding of the role of autophagy in a broader sense. As a conclusion, the results presented in this study could be used to understand the novel functions of autophagy in different taxa.

## ACKNOWLEDGEMENTS

We owe special thanks to Dr. Ronald E. Pearlman (York University, Toronto, Canada) for his critical reading of the manuscript. During peer-review period of our manuscript, The Nobel Prize in Physiology or Medicine 2016 was awarded to Dr. Yoshinori Ohsumi "for his discoveries of mechanisms for autophagy." On this opportunity, we would like to dedicate our paper to Dr. Ohsumi for introducing us to autophagy.

### Funding

The authors received no funding for this work.

### Competing Interests

The authors declare there are no competing interests.

### Author Contributions

- Erhan Aslan conceived and designed the experiments, performed the experiments, analyzed the data, contributed reagents/materials/analysis tools, wrote the paper, prepared figures and/or tables, reviewed drafts of the paper.
- Nurçin Küçükoğlu performed the experiments, analyzed the data, contributed reagents/materials/analysis tools, prepared figures and/or tables, reviewed drafts of the paper.
- Muhittin Arslanyolu contributed reagents/materials/analysis tools, reviewed drafts of the paper, gave scientific advise.

## Data Availability

The raw data has been supplied as a Supplementary File.

## Supplemental Information

Supplemental information for this article can be found online at http://dx.doi.org/10.7717/peerj.2878#supplemental-information.

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
