# Peer review of "A comparative in-silico analysis of autophagy proteins in ciliates"

_PeerJ, doi:10.7717/peerj.2878_

## Round 0.1 · original submission · Minor Revisions

Two experts in this field reviewed your manuscript and found that your findings are valuable for general readers. However, they also provide a number of criticisms. I found all of them to be reasonable, and suggest that you revise according to their comments. I hope you also find their comments are valuable to improve your manuscript.

Reviewer 1 ·

Basic reporting

The use of "lower" and "higher" eukaryote in the manuscript is confusing (and indeed these terms have no biological meaning). The authors state, for example, that autophagy is little understood in lower eukaryotes (in the abstract) but of course yeast is considered a lower eukaryote by people who use this term. These terms therefore confuse any evolutionary argument the authors are trying to make. Similarly, ciliates cannot be described as "an early branch of the eukaryotic clade" since there is nothing "early" about the branching of ciliates compared to other supergroups.
Better proofreading is needed. For example, line 107, 'elaborative' should be 'elaborate'; line 110, remove "have"or "share"; line 162, cerevisiae is spelled incorrectly. The term orthologs is used incorrectly in multiple places; for example; line 197: you cannot have two Atg18 orthologs to a single gene in another species - this is the definition of paralog. In places where statements are made about the properties of an autophagy-related protein, the authors should always say in what species the result was obtained, e.g., line 193. This can be extremely important. For example, lines 209/210, regarding the essential nature of the C-terminal domain in Atg1: is this result restricted to yeast and animals? If so, how do the authors rule out the possibility that this property of Atg1 is limited to Opisthokonts, in which case the ciliate Atg1 potential homologs aren't evolutionary outliers?
Some statements in the text appear contradicted in the figures. Line 234: except for Atg14, all the complex members are stated to be present in ciliates. But in figure 3B, VPS38 is marked as missing. The figure is confusing.
It is not clear what all the trees in the figures are meant to convey. For example, Fig 3C does not seem to contain any information not easily conveyed in the text. Moreover, the tree is unconvincing since yeast appears as basal to plants and animals. In figure 5D, the meaning of the dotted line is unclear. This tree should also be supplemented by one constructed using another algorithm, to support to placement of key nodes.
In figure 7, what is the logic of looking at the expression profiles of only a subset of the Tetrahymena/Paramecium ATG genes?

Experimental design

Whether this manuscript describes "primary research" is difficult to judge.
The heart of this manuscript consists of BLAST searches to identify ciliate homologs of autophagy-related genes. Unfortunately, the authors do not provide any numbers to back up their claims from these searches. Since there is no experimental work to back up the claims, it is essential that all data regarding the searches themselves be presented. Orthologs or paralogs can be demonstrated either by showing phylograms (which, in most cases, should be derived using at least 2 different methods, to show that critical nodes are reproducible; Maximum Likelihood and Bayesian would be best). If the authors instead want to rely on BLAST and reciprocal BLAST searches, they need to show the results (e.g., top 3-4 hits, including e-values) of both forward and reciprocal BLAST searches, for each homolog they discuss.
For cases in which the authors claim that no ciliate homologs exist, the authors should consider additional approaches. For example, they claim in lines 338-345 that ciliates lack a set of non-core ATG genes. However, several of these homologs (e.g., ATG20, 24) have previously been reported in ciliates, in a paper cited by the authors (Duszenko et al.). Some of the ciliate homologs may be impossible to identify using simple sequence-based searches, but be revealed by structure-prediction algorithms, e.g., hhpred.

Validity of the findings

Because of the shortcomings in data presentation described above in Experimental Design, it is not possible to judge the validity of the findings in general.

Additional comments

The authors are addressing a very interesting aspect of ciliate cell biology, with potentially broad implications for the evoution of autophagic mechanisms. Since they are relying solely on rather limited informatics-based approaches, they need to show the complete data to demonstrate the validity of their conclusions.

·

Basic reporting

The authors described the Atg proteins in ciliates genome. Using in silico analysis, the authors identified the absence of ATG1 gene in ciliate and ATG8 gene in Ichthyophthirius. The importance of different domains in Atg proteins and the functional roles in autophagy were discussed.

Experimental design

Used all the tools to do the analysis by in silico. Well designed

Validity of the findings

Checked the validity and found it is good

Additional comments

1) Line 32 and 56 autophagosome is written as organelle instead of structure.

2) What is the status of Atg8 in other system and why do the authors describe only in Ichthyophthirius?

3) The authors have done good literature study. In line 71 in the Introduction, few diseases to be added for Beclin-1 with references. Line 233 in the results, a reference to be added in the section.

4) An elaborate description of life cycle of Tetrahymena is not necessary.

5) In the mRNA expression analysis, gene names to be given along with the accession number in the first part of the results and not in the later part.

6) Spacing should be checked throughout the manuscript (example line 164 and 258).

---

## Round 0.2 · accepted · Accept

I find your rebuttals are satisfactory.